

# Effective population sizes and adaptive genetic variation in a captive bird population

Giridhar Athrey[1,2], Nikolas Faust[1], Anne-Sophie Charlotte Hieke[1] and I. Lehr Brisbin[3]

[1] Department of Poultry Science, Texas A&M University, College Station, TX, United States of America
[2] Faculty of Ecology and Evolutionary Biology, Texas A&M University, College Station, TX, United States of America
[3] Savannah River Ecology Lab, Aiken, SC, United States of America

## ABSTRACT

Captive populations are considered a key component of *ex situ* conservation programs. Research on multiple taxa has shown the differential success of maintaining demographic versus genetic stability and viability in captive populations. In typical captive populations, usually founded by few or related individuals, genetic diversity can be lost and inbreeding can accumulate rapidly, calling into question their ultimate utility for release into the wild. Furthermore, domestication selection for survival in captive conditions is another concern. Therefore, it is crucial to understand the dynamics of population sizes, particularly the effective population size, and genetic diversity at non-neutral and adaptive loci in captive populations. In this study, we assessed effective population sizes and genetic variation at both neutral microsatellite markers, as well as SNP variants from the MHC-B locus of a captive Red Junglefowl population. This population represents a rare instance of a population with a well-documented history in captivity, following a realistic scenario of chain-of-custody, unlike many captive lab populations. Our analyses, which included 27 individuals comprising the entirety of one captive population show very low neutral and adaptive genetic variation, as well as low effective sizes, which correspond with the known demographic history. Finally, our study also shows the divergent impacts of small effective size and inbreeding in captive populations on microsatellite versus adaptive genetic variation in the MHC-B locus. Our study provides insights into the difficulties of maintaining adaptive genetic variation in small captive populations.

# INTRODUCTION

## Genetics of captive populations

The management of demographic security and genetic diversity are among the central considerations in conservation (*Ralls & Ballou, 1986*). Captive populations have long played a key role in conservation as a management tool to ensure demographic security (*Hedrick, 1992*). The earliest work in conservation of endangered species focused on breeding in

Corresponding author
Giridhar Athrey,
giri.athrey@tamu.edu

captivity to increase population sizes and for eventual release back into the wild. However demographic security and the maintenance of genetic diversity can sometimes be at odds, as genetic diversity is derived from the effective population size ($N_e$). One issue in captive populations is the differential reproductive success of some breeding pairs that are better suited to captive conditions, leading to selection for survival in captive environments. This phenomenon can further reduce the $N_e$ due to founder effects (*Nei, Maruyama & Chakraborty, 1975*; *Newman & Pilson, 1997*). From the perspective of the reintroduction of the offspring of such captive breeders back into the wild, this may be a less than ideal situation, as these captive-adapted offspring may not carry the adaptive genetic variation required for survival in their natural habitat. Historically, captive individuals were raised for the genetic support of threatened wild populations, and to maintain genetic compatibility (prevent outbreeding depression) and genetic variation (ability to adapt to natural environment). However, these objectives can be materialized only if genetic diversity can be sustained, and if genetic drift and inbreeding can be limited. The relationship between population bottlenecks and reduction in effective population sizes and genetic diversity has been reported from various vertebrate taxa (*Leberg, 1992*; *Athrey et al., 2011*; *Athrey et al., 2012*; *Lovatt & Hoelzel, 2014*).

The loss of genetic diversity due to drift and inbreeding in captive populations has also been well documented (*Willoughby et al., 2015*; *Willoughby et al., 2017*) and the drawbacks of using closed captive breeding populations for reintroduction have been described (*Lynch & O'Hely, 2001*). These range from the rapid decline of fitness upon reintroduction (*Araki, Cooper & Blouin, 2007*) to a range of consequences for genetic diversity (*Witzenberger & Hochkirch, 2011*). *Witzenberger & Hochkirch (2011)* also suggest the need for assessment of genetic diversity before *ex situ* conservation programs. The consensus of these studies suggest that considerations of genetic diversity of captive populations may be as crucial as maintaining demographic stability, but is perhaps much more challenging than the maintenance of demographic stability, because mating among relatives is not entirely avoidable. While some studies have focused on assessing genetic diversity, in many instances the history of the founding individuals is not well known, or the studbook data may be incomplete (*Witzenberger & Hochkirch, 2011*). Additionally, inbreeding estimates may assume that founders are unrelated (*Ruiz-López et al., 2009*). For example, *Alcaide et al. (2010)* compared wild versus captive and reintroduced Kestrel, but in that instance, it is not clear how many generations the kestrels had bred in captivity. In another case study using experimental captive breeding, *Willoughby et al. (2017)* report on white-footed mice populations that were bred in captivity for 20 generations, and were able to observe mitigating effects on genetic diversity using a very deliberate mating protocol (mean-kinship method). However, in many instances, there may be no choice available as to the source of individuals obtained and used in captive breeding programs. Furthermore, in extreme cases, small numbers of breeding individuals may have to be maintained in captivity for decades until suitable habitat and molecular/technological tools to enable *ex situ* applications become applicable. From that perspective, it is essential to have realistic estimates of how much genetic diversity is maintained, and whether estimates of genetic

diversity at neutral markers are informative about variation at loci crucial for survival in the wild.

In this study, we used a captive population of the Red Junglefowl (*Gallus gallus murghi*) to characterize the impacts of un-managed captive breeding over a period of 57 years (38–57 generations). The study population called the Richardson Strain (RRJF) has a well-documented history since its importation to the United States in 1961. The history of this population is illustrated as a flowchart in Fig. 1. We investigated the consequences of captive breeding on genetic diversity at neutral and adaptive genetic markers experienced by this population since the early 1960s.

## Demographic history of Richardson's Red Junglefowl (RRJF)

The study population of Red Junglefowl (RJF) was initially imported from Northern India in 1961 as part of the US Fish and Wildlife's Foreign Game Introduction Program (FGIP) and documented in USFWS Special Reports (*Bump & Bohl, 1964*). More recently, the history of the population in the United States was documented in detail in the graduate thesis of Tomas Condon (*Condon, 2012*). Briefly, the birds that were imported to the United States were captured in the Indian states of Bihar and Uttar Pradesh (now Uttarakhand), between 1959–1961. A total of 119 survived the capture and shipping to the United States. These birds were placed in hatcheries across nine different states, and over the next decade, approximately 10,000 birds were released into US habitats for hunting, until the FGIP program ended. Over the next four decades, almost all the descendant RJF: (a) failed to survive in their new habitats, (b) became incorporated into private aviaries, or (c) remained in small breeding populations maintained by dedicated researchers and aviculturists. All the surviving RJF of known ancestry (estimated around 100–150) were derived from a small number of birds. These birds have a well documented chain of custody. In 1969, I. Lehr Brisbin (of the Savannah River Ecology Lab) received five birds from the South Carolina Department of Wildlife Resources, to which 26 were added in 1970. By 1971, this flock numbered only eight adult birds (four of each sex). While more hatchlings were added to this in mid-1971, the population remained at under 15 individuals, until they were transferred to Mr. Isaac Richardson of Alabama. Mr. Richardson started his colony with 12 individuals (8 males). Between 1972–2010, the colony was sustained by Mr. Richardson, which later became known as the Richardson's Red Junglefowl (RRJF). Over this time, the population expanded and was distributed to a few others, but at any time, the breeding colony never exceeded 20 individuals (*Condon, 2012*). Based on behavioral and morphological studies of Red Junglefowl, it has been argued that the RRJF represent a pure and unique population of Red Junglefowl (*Brisbin & Peterson, 2007*), whose preservation is critical in the face of hybridization with domestic chickens in the native range of Junglefowl (*Peterson & Brisbin, 1998*; *Brisbin & Peterson, 2007*).

The research colony maintained at Texas A&M University (TAMU) is one of the two known remaining populations that trace their ancestry back to the survivors of Mr. Richardson's flock. The TAMU colony was established with two males and three females and expanded to 27 individuals at the time of the study. All available individuals were included in this present study.

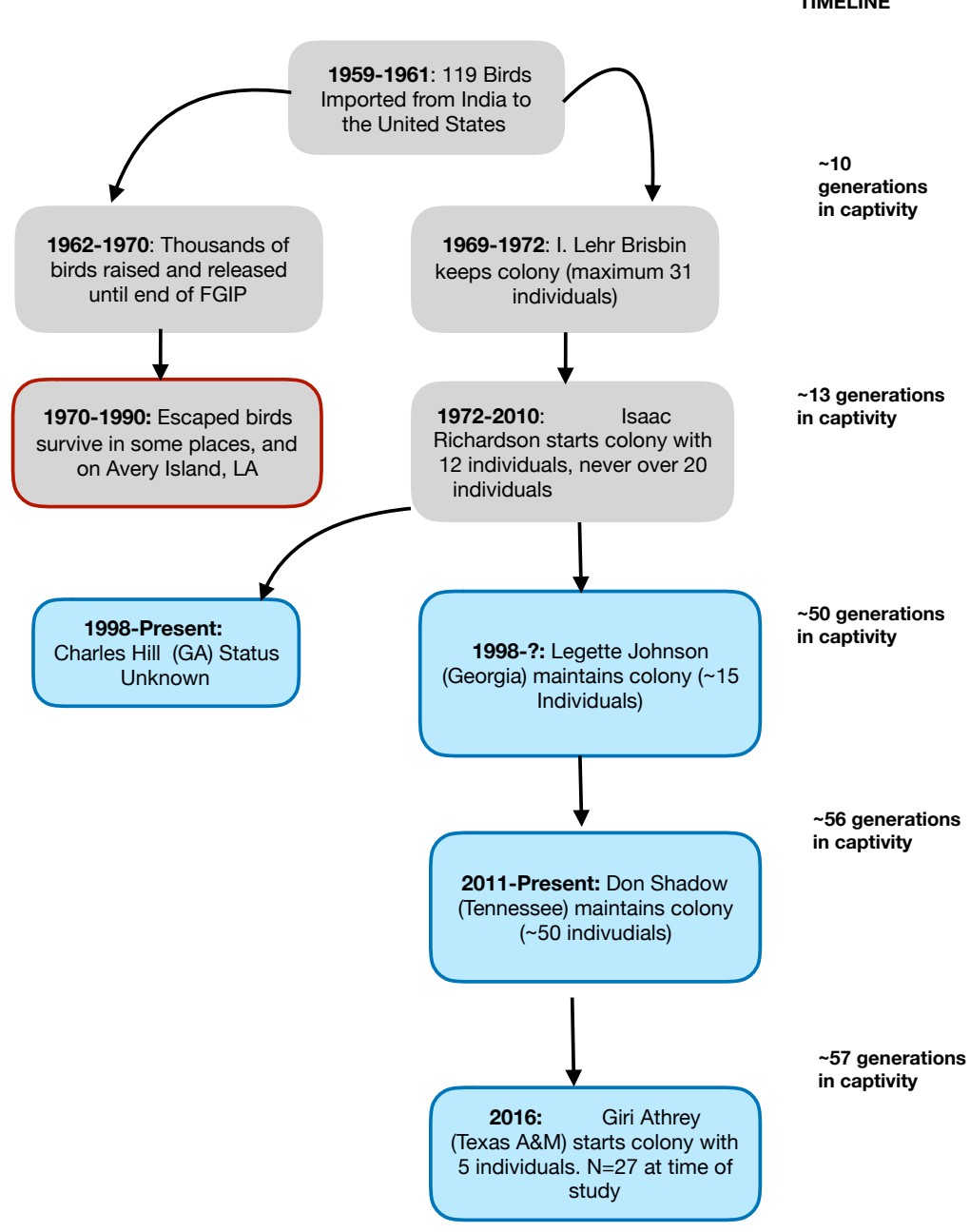

**TIMELINE**

**1959-1961**: 119 Birds Imported from India to the United States

~10 generations in captivity

**1962-1970**: Thousands of birds raised and released until end of FGIP

**1969-1972**: I. Lehr Brisbin keeps colony (maximum 31 individuals)

**1970-1990**: Escaped birds survive in some places, and on Avery Island, LA

~13 generations in captivity

**1972-2010**: Isaac Richardson starts colony with 12 individuals, never over 20 individuals

**1998-Present**: Charles Hill (GA) Status Unknown

~50 generations in captivity

**1998-?**: Legette Johnson (Georgia) maintains colony (~15 Individuals)

~56 generations in captivity

**2011-Present**: Don Shadow (Tennessee) maintains colony (~50 indivudials)

~57 generations in captivity

**2016**: Giri Athrey (Texas A&M) starts colony with 5 individuals. N=27 at time of study

**Figure 1** **A flowchart showing the known demographic history of the Richardson's Red Junglefowl.** A graphical representation of the known population history of the Richardson's Red Junglefowl population in captivity and chain of custody. The red rimmed box denotes released populations that ultimately failed to take hold, and experienced extinction in their new habitat. The blue shaded boxes show the history of the populations that trace back to Isaac Richardson's flock.

## Genetic diversity and effective population sizes

As the management of genetic diversity is one of the primary intentions of captive breeding programs, it is important to assess the consequences of captive breeding over multiple generations on genetic diversity. Other authors have shown reductions in genetic diversity in captive populations, and these typically result from differential survival and increased genetic drift, but also a form of (unintentional) domestication selection (*Briscoe et al., 1992*; *Montgomery et al., 2010*). In these instances, it is important to characterize what the effective population size is, as this one parameter can drive various other genetic parameter estimates. The effective population size ($N_e$) can be defined as the size of an ideal population that experiences the same amount of genetic drift or inbreeding as the actual (census) population (*Wright, 1932*; *Wright, 1948*). A number of approaches have been developed to estimate $N_e$—for example, temporal estimates based on two or more samples, methods based on linkage disequilibrium, and coalescent approaches (*Wang, Santiago & Caballero, 2016*). Due to our access to a single point in time, as well as our interest in understanding recent effective population size, we focused only on methods for contemporary or recent $N_e$ estimates. In this study, we evaluated genetic diversity, estimated the effective population size ($N_e$), and analyzed linkage disequilibrium using two types of markers—namely microsatellite markers and SNP markers. The SNP markers assayed genetic diversity along chromosome 16 (details in 'Methods'), a microchromosome which contains both the MHC class I and class II genes, in addition to olfactory receptors and scavenger proteins (*Miller & Taylor, 2016*). The MHC (major histocompatibility complex) is a collection of genes that form a crucial part of vertebrate adaptive immunity, and in chicken they have been demonstrated to be important in resistance against diseases such as Marek's Disease (*Shiina, Hosomichi & Hanzawa, 2006*; *Miller & Taylor, 2016*).

# MATERIAL AND METHODS

## Study population

A total of 27 birds bred in captivity at the Texas A&M Poultry Research Center were genotyped in this study. The population represents three generations. The parental cohort was comprised of two males and three females in January 2016 and had expanded to 27 individuals by the second half of 2017. At the time of sampling, the population included original parents (used to establish colony), in addition to two subsequent generations that were hatched in Texas. All individuals available were included in this study.

## Molecular methods

Venal blood was sampled from all individuals by puncturing the brachial vein and stored in the Tris-EDTA based stabilization solution, Longmire Buffer, until further processing (*Longmire et al., 1997*). Birds were sampled in accordance with protocols approved by the Texas A&M Institutional Animal Care and Use Committee (IACUC 2016-0065). Genomic DNA was isolated from whole blood using the DNeasy Blood and Tissue kit (Qiagen Inc, Hilden, Germany). For microsatellite genotyping, a total of 18 microsatellite markers were selected from a list of known chicken microsatellite loci (*Tadano et al., 2007*) and screened for the presence of polymorphic loci. The selection of loci was based on our intention to

characterize marker variation across the genome. The 18 loci screened were each from a different chromosome, and were picked based on their reported polymorphism. Of the screened loci, three loci were monomorphic in our study population, or were not amplified by PCR consistently and were not used further (full list and primer sequences presented in Table S1). To generate genotype data from microsatellite loci, we used the fluorescently labeled M13 method (*Schuelke, 2000*), which allows greater flexibility of fluorescent dye tagging for multiplexing. PCR thermoprofile was optimized based on published $T_m$ for each primer pair (and considering M13 sequence), and amplification was performed in a 25 μl reaction using NEB Taq Polymerase with reaction buffer (New England Biolabs, Ipswich, MA, USA) and NEB dNTPs (New England Biolabs). PCR amplification was performed on an Eppendorf Mastercycler Pro thermal cycler (Eppendorf, Hauppauge, NY, USA), running an initial denaturation at 95 °C for 5 min, followed by 35 cycles of denaturation (95 °C, 30 s), annealing ($T_m$, 10 s), and extension (72 °C, 45 s), and a final extension for 10 min. Each microsatellite locus was amplified and then pooled for multiplex genotyping based on combinations of fluorescently labeled probes. Multiplex genotyping was performed on the ABI 3730 capillary analyzer at the DNA sequencing facility on Science Hill (Yale University, New Haven, CT, USA) for fragment analysis. For SNP analyses, equimolar DNA isolates were submitted to the molecular genetics lab at Hy-Line International for genotyping using a custom SNP panel (*Fulton et al., 2016b*; *Nguyen-Phuc, Fulton & Berres, 2016*). The SNP panel was developed as an assay for the Major Histocompatibility Complex- B locus (MHC-B), based on the KASP$^{TM}$ chemistry (LGC group, Teddington, UK) where each allele is identified by its fluorescent label (VIC or FAM). A total of 90 known SNP loci are genotyped using this panel and covers the region between basepairs 30,189 to 240,933 on chromosome 16. While microsatellite loci came from multiple chromosomes (Table S1), all the SNP loci are from a single gene-rich microchromosome, and as is typical for SNP datasets, the markers represent a mix of intronic/intergenic (putatively neutral) and coding regions.

## Genetic analyses

The raw capillary electrophoresis data was downloaded into the software Geneious (Biomatters, New Zealand) and analyzed using the microsatellite plugin. Allele bins, based on peak data from across all samples, were created for each locus. Following this, every individual biological sample was processed through the peak calling step. Once the automatic peak calls were obtained, every allele call was manually verified to check for errors. The final allele calls were exported into the MS Excel-based tool, GENALEX v6.5 (*Peakall & Smouse, 2006*). The data were then exported to the GENEPOP format for estimation of the effective population size. The SNP calls were generated (by Hy-Line International) using the Kraken software (LGC group, UK). IUPAC base calls were first re-coded manually into 2-letter genotypes, and then imported into GENALEX and recoded into a numeric format for further analysis. Estimation of expected and unbiased expected heterozygosity ($H_e$ and $uH_e$), and measures of the inbreeding coefficient $F_{IS}$ were both calculated within GENALEX, based on 999 permutations of the data. These parameters were estimated for both the microsatellite and SNP data independently.

## Effective population size

Next, to estimate the effective population size of this captive-bred population, we used three different estimators that are known to estimate contemporary or recent effective population sizes—namely the heterozygosity excess (HE) method (*Zhdanova & Pudovkin, 2008*; *Pudovkin, Zhdanova & Hedgecock, 2010*), the linkage disequilibrium (LD) method (*Waples & Do, 2008*) and the molecular coancestry (MC) method (*Nomura, 2008*). All three $N_e$ estimates were generated using the software program NeEstimator, version 2 (*Do et al., 2014*). These estimators are expected to estimate the population size based on shared alleles (MC) or the signal of genetic drift—either due to allele frequency differences among parents (HE) or due to the linkage disequilibrium among markers (LD). In this case, we also know which assumptions were violated with some certainty. For example, this captive population has been closed to immigration and not subject to artificial selection (except unintentional domestication selection in captivity), but the assumption of random mating is likely to be violated, as we expect this population to have a high frequency of mating among relatives over the last five decades. While it is known that most real natural populations may violate one or more assumptions of an ideal population, the departure from random mating is expected to result in underestimates of $N_e$ (*Waples, Antao & Luikart, 2014*). Similarly, the molecular coancestry estimate is expected to be biased downward in inbred populations (*Nomura, 2008*). Considering the differential consequences of these assumptions for our captive population, we estimated $N_e$ using the three methods mentioned above. The analyses were carried out separately for the microsatellite data, and for the SNP data. Furthermore, we were particularly interested in determining the $N_e$ as obtained from markers from a single chromosome. By definition, a chromosome is a single linkage group (*Groenen et al., 2000*; *Wright et al., 2010*), and linkage of loci over a single chromosome can be expected to be stabler over evolutionary time, compared to markers sampled from across the genome. However, estimates of contemporary $N_e$ are expected to be biased downward by the length of the chromosome, and also the number of chromosomes (*Waples, Larson & Waples, 2016*) primarily due to physically linked loci, and recombination frequencies being lower on short chromosomes. To our knowledge, this is the first study where a high density of markers from a single chromosome was used to explore the population history. For estimation of $N_e$, we used the phased genotypes from each individual as input (details below).

## Haplotype and linkage analyses

To understand the effects of captive breeding on the number and structure of unique haplotypes in the captive population, we first phased the SNP genotype data to identify the unique haplotypes present in the population. We used the program PHASE v2 (*Stephens, Smith & Donnelly, 2001*) to phase the SNP loci into unique haplotype sequences. We performed phasing and estimation of haplotype frequencies from five replicate runs of the program PHASE. Each replicate run started with a different seed and was comprised of 1,000 iterations, with a thinning interval of five and a burn-in interval of 100. Following the completion of these runs, the number of unique haplotypes and their frequencies were recorded and checked across replicate runs. As the number of haplotypes and

**Table 1  Summary of genetic diversity in the study population based on microsatellite and SNP loci.**
Estimates of genetic diversity for both microsatellite and SNP datasets are presented. For each marker *type*, genetic diversity measures are shown along with their standard errors. The columns present the number of alleles (Na), effective number of alleles (Nae), observed heterozygosity (Ho), expected heterozygosity (He), the unbiased expected heterozygosity (uHe), and the fixation index F.

| Marker Type | | Na | Nae | Ho | He | uHe | F |
|---|---|---|---|---|---|---|---|
| **Microsatellite** | Mean | 2.33 | 1.74 | 0.36 | 0.37 | 0.38 | 0.07 |
| | SE | 0.16 | 0.14 | 0.06 | 0.05 | 0.05 | 0.08 |
| **SNP** | Mean | 1.38 | 1.13 | 0.11 | 0.1 | 0.1 | −0.14 |
| | SE | 0.05 | 0.02 | 0.01 | 0.01 | 0.01 | 0.02 |

their frequencies was very consistent across runs, we determined that increasing the run length was not necessary (*Stephens, Smith & Donnelly, 2001*). The list of unique haplotypes was then used to construct a haplotype network using the 'APE' package on the R statistical platform (*Paradis, Claude & Strimmer, 2004*; *Popescu, Huber & Paradis, 2012*). This approach constructs a matrix of Kimura-2 parameter genetic distance (*Kimura, 1980*) based on nucleotide differences among the haplotypes and plots a network representing differences among haplotypes. We also estimated nucleotide diversity ($\pi$), haplotype diversity, and estimate of Tajima's D, using the R package PEGAS (*Paradis, 2010*).

Finally, to examine linkage structure across chromosome 16 based on the 90 SNP loci, we used the program HaploView (*Barrett et al., 2005*). This program summarizes estimated Minor Allele Frequency (MAF) for each locus and generates haplotype and linkage information for locus pairs, which are then represented in a graphical format. Higher values of linkage disequilibrium (LD) suggest stronger association among loci, and in turn, indicate low recombination between loci.

## RESULTS

### Genetic diversity

For the microsatellite data, 93.3% of the loci were polymorphic, and the three estimates of heterozygosity based on microsatellite markers were very similar to each other (observed = 0.359, expected = 0.371, unbiased = 0.379). For the SNP dataset, only 37.8% of the loci were polymorphic, which translated into heterozygosity estimates of 0.107, 0.096, and 0.098 (observed, estimated, and unbiased, respectively, Table 1). Estimates of F, the fixation index/inbreeding coefficient, were also divergent between the microsatellite (0.065) and SNP datasets (−0.137). Positive values of F are typically indicative of more significant inbreeding than expected, whereas negative values suggest more outbreeding than expected. This latter result is potentially a consequence of the low observed and expected heterozygosities for the SNP dataset, arising from the small proportion of SNP loci that were polymorphic.

### Unique haplotypes and linkage

As expected, analysis of linkage disequilibrium at MHC-linked SNP loci on chromosome 16 revealed very high D′ (linkage disequilibrium) values across all pairwise comparisons. D′ was estimated to have a value of 1, indicating little or no recombination between marker

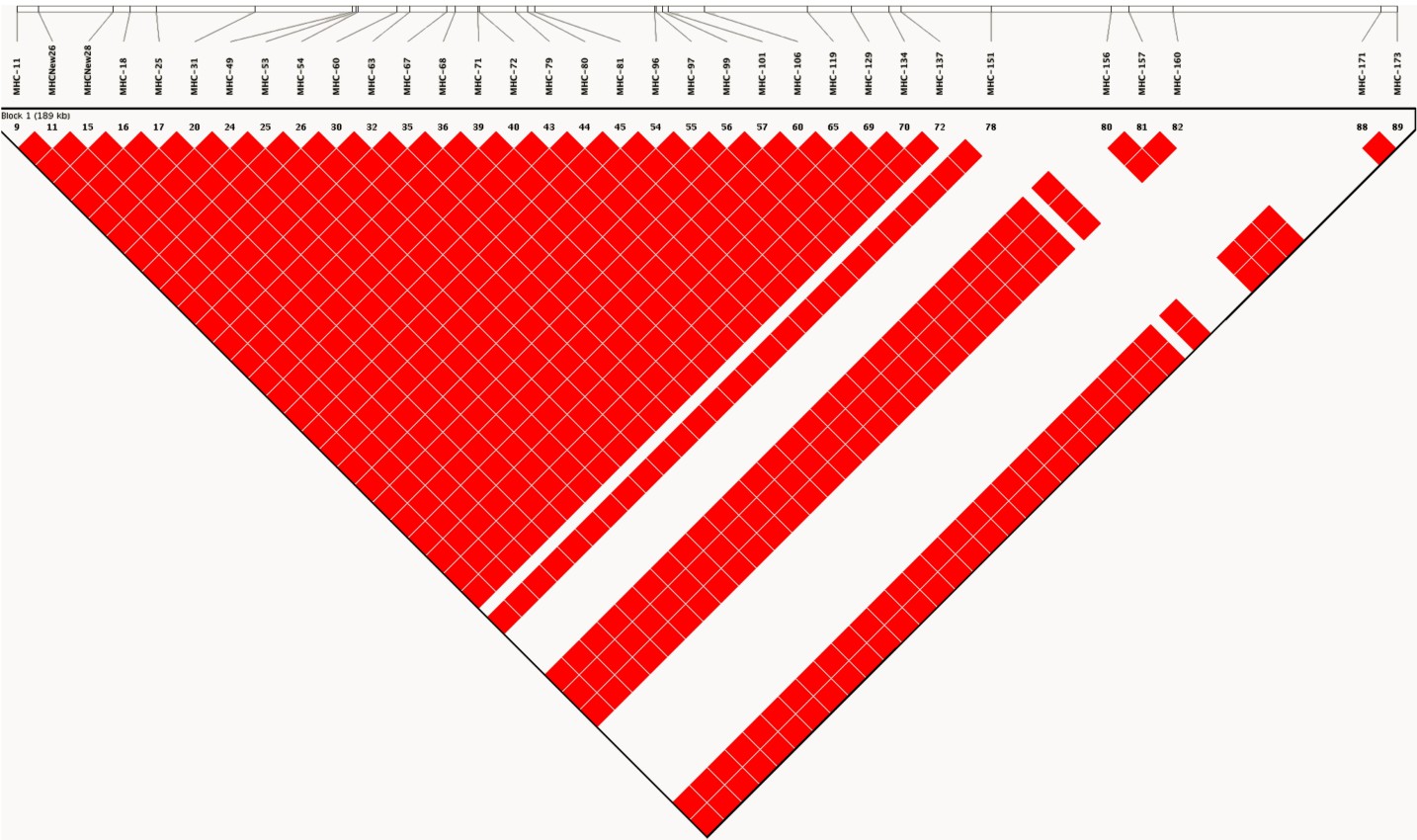

**Figure 2** **LD plot generated based on recombination frequencies across the MHC-B locus as characterized by the SNP panel.** A linkage map based on the 90 SNP loci on chromosome 16, generated from genotype data for the study population. The map shows pairwise estimates of linkage disequilibrium between markers. Red colored blocks suggest high linkage disequilibrium (D′) values of 1, implying little to no recombination between markers.

pairs (Fig. 2). Such high values indicate that as a consequence of captive breeding in a small founding population, individuals in this population have inherited virtually identical chromosomes (chromosome 16 at least), a phenomenon that would be indistinguishable from the absence of recombination.

Analysis of haplotypes in the study population using the program PHASE revealed five unique haplotypes (Fig. 3), and of these only three were found at frequencies higher than 10%, with the most common haplotype represented in 70% of individuals (haplotype 1). The next two most common haplotypes were found at approximately 14% frequency each. These haplotypes suggest at best three unique haplotype lineages contributing to the population and presumably do not reflect novel haplotypes emerged by mutations since the initial population contraction. Estimate of nucleotide diversity ($\pi$) was 0.009, whereas the haplotype diversity was 0.43. Finally, we found a significantly negative Tajima's D based on the haplotype sequences $-3.19$ ($P < 0.01$), suggesting a genetic bottleneck (or directional selection). Comparison of nucleotide diversity and haplotypes at MHC loci from wild jungefowl and domestic chicken from other studies are shown in Table 2.

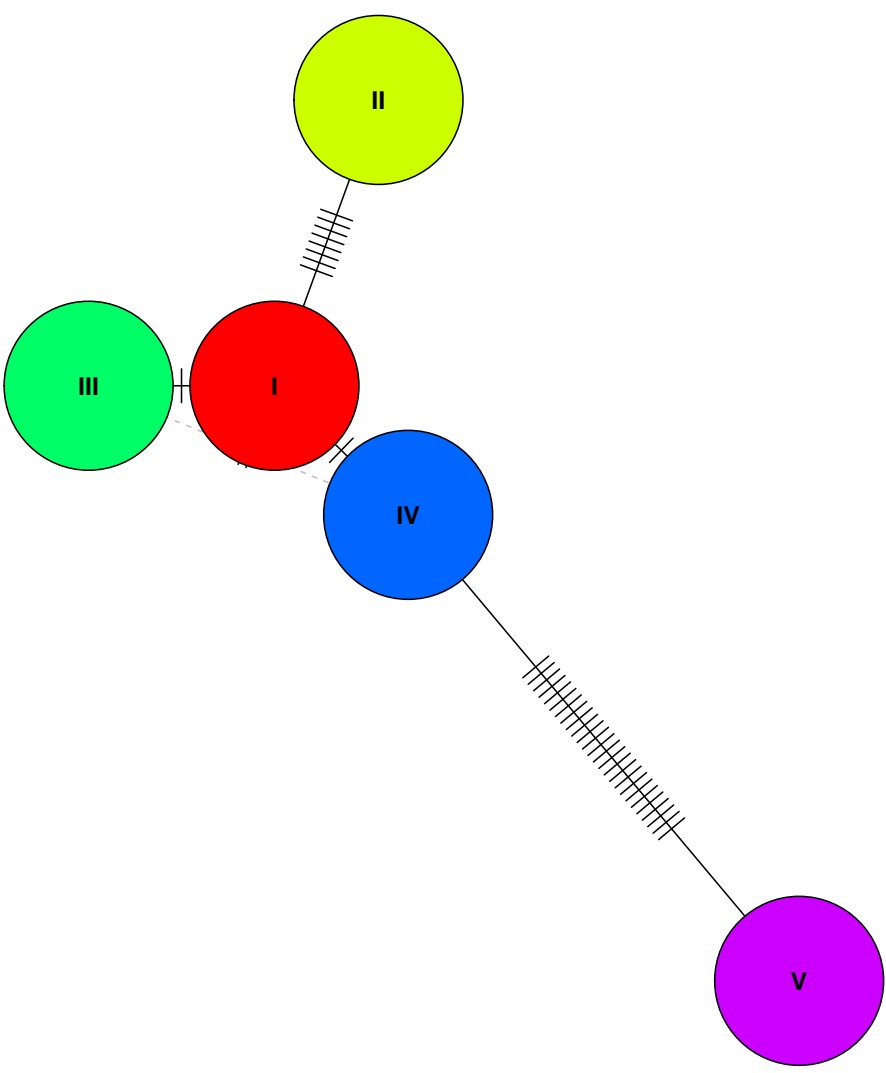

**Figure 3  A haplotype network based on genetic distances between the five haplotypes identified in the study population.** A haplotype network of the the five (phased) haplotypes detected in the study population. Only three of the five unique haplotypes were found at a frequency of over 10% in the population (I, III, and V).

## Effective population size estimates

The $N_e$ estimates were less than 10 for the microsatellite marker dataset, except for the heterozygote excess method (Table 3). For the single chromosome SNP dataset, both the LD and MC estimates were 1 or lower with narrow 95% intervals, whereas the $H_E$ estimator was in the low single digits ($N_{eHE} = 4.8$, 95% CI = 3.1–11). If these estimates are compared against the total number of individuals in the local population (Texas A&M), the $N_e/N_c$ ratios range from about 0.14–0.27 for microsatellite markers, whereas they range from 0.007–0.17 for the SNP based estimates.

**Table 2  Comparison of SNP based genetic diversity measures between study population, and other junglefowl and chicken studies.** Estimates of nucleotide diversity, haplotype number and diversity and Tajima's D in captive study population, compared against wild junglefowl populations and domestic chicken. All the studies compared here used the same high-density SNP panel covering the MHC-B locus on Chromosome 16.

| Type | N | Nucleotide diversity | Haplotype diversity | Unique haplotypes | Tajima's D | Source |
|---|---|---|---|---|---|---|
| Red Junglefowl | 27 | 0.009 | 43% | 5 | −3.191 | This study |
| Red Junglefowl (5 wild populations) | 199 | 0.28 (avg) | 99% (avg) | 310 | 2.1 | *Nguyen-Phuc, Fulton & Berres (2016)* |
| Finnish Landrace Chicken (12 populations) | 195 | Not available | Not Available | 36 | Not Available | *Fulton et al. (2017)* |
| Domestic chicken (17 breeds) | 1,351 | Not available | Not available | 86 | Not Available | *Fulton et al. (2016a)* |
| Domestic chicken (5 breeds) | 112 | 0.05 | 74% | 31 | −2.12 | G Athrey, 2018, unpublished data |

**Table 3  Effective population size estimates for microsatellite and SNP markers in the study population.** Estimates of effective population size Ne are presented for microsatellite and SNP datasets. Ne estimates from three estimators are shown along with their respective 95% confidence intervals. Ne was estimated using the Heterozygote Excess (NeHE), Linkage Disequilibrium (NeLD), and the Molecular Coancestry (NeMC) methods.

| | NeHE | 95% CI | NeLD | 95% CI | NeMC | 95% CI |
|---|---|---|---|---|---|---|
| Microsatellite | inf | 7.3-inf | 7.3 | 3.6–12.5 | 4 | 1.1–8.8 |
| SNP | 4.8 | 3.1–11 | 0.2 | 0.1–0.2 | 1 | 1–1 |

## DISCUSSION

Our study generated valuable new understanding of genetic diversity and effective population sizes in populations that have persisted in captivity for a fairly small number of generations (<60). Additionally, our comparison of genome wide neutral microsatellite diversity, and high-density SNP survey of adaptive genetic loci revealed important differences in how these marker types respond to population bottlenecks, and how that is reflected in the estimates of effective population size; while microsatellite-based genetic diversity estimates were low, they were, nonetheless, higher than SNP-based estimates for the MHC-linked region. These differences were also reflected in the population histories revealed by these markers; effective size estimates were extremely low for the MHC-linked SNP loci. The disparity in estimates between the neutral and adaptive marker types is potentially concerning from the perspective of captive population based *ex situ* conservation programs, and also has implications for endangered wild populations with small numbers of breeding individuals. One important suggestion is that estimates of functional genetic diversity and effective population sizes are overestimated by microsatellite loci.

### Genetic diversity

We found relatively low genetic diversity based on heterozygosity, among both marker types, but estimates of heterozygosity were higher based on microsatellite markers compared to MHC-linked SNP loci. This is not a surprising result, as microsatellites may not be highly correlated with underlying genomic diversity (*Väli et al., 2008*), for example, due

to ascertainment bias for polymorphic loci. *Tokarska et al. (2009)* have also shown that in species or populations with low genetic diversity, heterozygous SNP loci may be more informative about population history and structure. In our case, the MHC-linked SNP marker panel included loci from both coding and noncoding regions, but all these markers came from a single chromosome.

While it is difficult in our study to determine how the MHC-linked SNP loci may be representative of whole genome diversity, in this study population we expect that chromosome 16 would be experiencing selection or drift in ways that are consistent with the rest of the genome. The MHC is a set of important adaptive loci (*Flajnik & Kasahara, 2001*), and in natural populations they may be expected to experience selection pressures distinct from the rest of the genome (*Sutton et al., 2011*; *Strand et al., 2012*; *Oliver & Piertney, 2012*). In chicken, the MHC loci have been shown to be important for adaptive immune responses against a number of infectious diseases, and hence is a target of both natural and artificial selection (*Kroemer et al., 1990*; *Kaufman & Wallny, 1996*; *Shiina et al., 2007*; *Miller & Taylor, 2016*). But, in this captive population with a history of small population size, and high degree of nonrandom mating, we expect drift to be a predominant evolutionary force. Furthermore, selection prior to a population bottleneck is expected to result in a disproportionate loss of diversity at MHC loci (*Sutton et al., 2011*). The recorded history does not show if any major selection events occurred immediately prior to the FGIP importation, but the breeding and release programs between 1962-1970 do not suggest any disease related mortality events (*Bump & Bohl, 1964*; *Condon, 2012*). The demographic history following arrival in the United States is suggestive mainly of random genetic drift. Hence we believe that the measures of diversity based on the MHC-liked SNP panel to be reflective of genome-wide patterns of diversity (both adaptive and neutral), while the microsatellite-based estimates are likely to represent genome-wide neutral marker diversity alone. If this supposition is correct, then assessing and, perhaps managing, genetic diversity at loci that underpin adaptive traits would be as important as assessing neutral genetic diversity, if not more so.

The second conclusion of the comparison between microsatellite markers and SNP markers is the potential implications of such disparity for other endangered or threatened species of conservation interest. Several studies have used microsatellite markers to assess genetic variation following population contractions (*Bouzat, Cheng & Lewin, 1998*; *Wisely et al., 2002*; *Johnson & Dunn, 2006*; *Athrey et al., 2011*; *Hammerly, Morrow & Johnson, 2013*). Due to the relatively low cost and accessibility of microsatellite loci, the usage of these markers has become ubiquitous in population and conservation genetics, and in the assessment of genetic diversity. If microsatellite-based measures of genetic diversity or effective population sizes are higher than SNP-based genetic diversity, as observed in the current study, it would be potentially concerning from the perspective of measuring the genetic viability of wild populations.

## Effective size estimates

We found low (single digit) $N_e$ estimates across all estimators. On the one hand this is consistent with the known demographic history of few breeding individuals at any given

time, on the other hand, low estimates may arise from small sample size or other conditions. We expect $N_e$ estimates to be biased downward due to the small number of individuals in this study, but these sample sizes are not unusual in endangered populations. Several of the $N_e$ estimators expect biased estimates at sample sizes under $N = 50$. However, in this study, 100% of individuals were sampled. Another source of bias, especially for the LD method, as reported by *Waples, Larson & Waples (2016)*, is the number and length of chromosomes. In our case, the SNP data were generated from chromosome 16—one of the smaller micro-chromosomes, which has an estimated length of 48–80 cM (*Burt & Cheng, 1998*; *Groenen et al., 2000*). We expect both the smaller sample size and the use of single chromosome SNP markers to be contributing to low $N_e$ estimates using the LD, and perhaps the MC method.

Given the known history of the study population, the low $N_e$ estimates were not unexpected, but the differences between the microsatellite markers from across the genome, and the single chromosome SNP markers were notable, and reveal different population histories. Both the MC and the LD methods estimated $N_e$ to be equal to or lower than one, based on the SNP dataset. Whether these differences are the result of different recombination frequencies for marker types or actual biological lineages for unique haplotypes is challenging to differentiate. Our finding of a low number of MHC haplotypes in this population suggests that fewer unique haplotypes to be driving the observed pattern. Due to our exact knowledge of the census size, the $N_e/N_c$ ratios are informative about both the processes of marker inheritance, as well as the performance of the estimators themselves. Secondly, all the three estimators used in this study are expected to represent the previous or recent parental generations. Therefore, for both the LD and MC methods, we believe that the estimates obtained here represent the founding event of the Texas population in 2016. On the other hand, as changes in heterozygosity occur more gradually over time, and the accumulation of inbreeding may take several generations, the HE estimator may be more representative of a founder event bounded by the initial founding of the US populations in 1961. However, as our analyses of haplotypes and recombination frequencies showed, all the MHC-linked SNP loci are in very high linkage disequilibrium. Such lack of independence among loci in the entire population will be indistinguishable from an effective size of one, which is what we observe here. Our results point to one of the limitations of LD methods in populations such as the study population, where the founders represent only five unique haplotypes. Secondly, the MHC region is inherited as haplotypes instead of segregating loci along the region (*Hosomichi et al., 2008*), which can also bias estimates of LDNe.

Finally, the number of microsatellite loci and the low diversity at these loci might be another source of bias in the estimates. Even though we chose microsatellite loci that were reported to be highly polymorphic, these loci had low diversity in the study population. While *Antao, Pérez-Figueroa & Luikart (2011)* reported that sample size is more crucial for detection of population size declines (especially with LD methods), in our case we sampled the entire census population size. While these methods are ideally suited for estimating $N_e$, when the true $N_e$ is low (<100), real world scenarios such as our study population (both the census and effective sizes are low) pose challenges for the application and interpretation

of $N_e$ estimates, and show the need for development of methods to assess and correct for bias in such situations.

## Haplotype diversity and linkage

We observed only three major haplotypes in the population, and it will be valuable to understand how these haplotypes are maintained in subsequent generations. We believe that the low haplotype diversity is a consequence of the history of captive breeding among a small number of individuals, rather than a reflection of the source populations. Haplotype diversity in other wild junglefowl populations has been reported to be quite high. For example, *Nguyen-Phuc, Fulton & Berres (2016)* reported 313 unique haplotypes from 199 wild-caught individuals from Vietnam. This suggests that the progenitors of the RRJF are likely to have come from a similarly diverse population, which has since lost genetic diversity in captivity. It is also worth noting that the genetic bottleneck and the resulting reduction in genetic diversity in only 50 years perhaps mirrors the early genetic history during selection for domestication, except for periodic gene flow from wild individuals. It would appear that such immigration was necessary to maintain the high levels of genetic diversity that is found in domestic chickens today (*Granevitze et al., 2007*; *Fulton et al., 2016b*). We observed high linkage disequilibrium among loci in the SNP dataset—an estimate that is known to be inflated when using small sample sizes, and very likely to be contributing to the high D′ values observed (*Ardlie, Kruglyak & Seielstad, 2002*; *England et al., 2006*). Taken together with the reduced haplotype diversity in this population, however, the linkage disequilibrium estimates are biologically plausible. Selection for tameness and success in captive environments is inevitable in captive breeding populations (*Frankham et al., 1986*; *Briscoe et al., 1992*; *Woodworth et al., 2002*). The genetic consequences of domestication have received much attention in recent years and the genetic architecture of traits—mainly driven by linkage and pleiotropy are considered to be crucial in the expression of specific phenotypes in domesticated varieties. For example, *Wright et al. (2010)* showed strong linkage blocks, as well as low heterozygosity regions associated with selective sweeps in domesticated varieties of chicken. The haplotype diversity and linkage patterns we observed in our captive population appear to be consistent with what might occur early during domestication, if a small number of individuals are selected. The high genetic diversity found in domestic chicken breeds and wild junglefowl populations, and also at the MHC-linked SNP loci (*Fulton et al., 2016a*; *Fulton et al., 2017*; *Nguyen-Phuc, Fulton & Berres, 2016*) would suggest that domestication of chicken would have included frequent immigration over several tens of generations. One glimpse at this is found in the estimates of Tajima's D from Wild Junglefowl (*Nguyen-Phuc, Fulton & Berres, 2016*) versus the captive junglefowl population (this study). Wild junglefowl populations show positive values of Tajima's D, suggesting diversifying or balancing selection (*Tajima, 1989*), whereas both the domestic chicken and captive junglefowl show negative Tajima's D values (Table 2). While we cannot distinguish between the effects of directional selection and bottleneck in our study, the latter is expected given the known history of this population. The major shift in diversity from wild to captive populations within 60 generations is intriguing, especially in the context of domestication selection.

## CONCLUSIONS

Our study showed low genetic diversity and small effective population sizes in a captive breeding bird population. Furthermore, our study was able to relate effective population sizes to the known demographic history of the captive population. Given this history, the low effective population size estimates we found are not surprising. Furthermore, the low genetic diversity, and haplotype diversity both indicate that these populations are indeed pure, and unmixed with domestic varieties, since their initial importation to the United States. Our study shows the rapid loss of genetic diversity in captive populations in only a few generations (50–57 generations since importation), which can be particularly concerning from the standpoint of *ex situ* conservation goals starting with captive populations. Finally, we showed how captive breeding populations, even in the absence of any intentional selection, can affect linkage structure across a set of adaptive genetic loci, and potentially reduce the fitness and adaptability of captive-born individuals upon release back into the wild.

## ACKNOWLEDGEMENTS

The authors thank Dr. Town Peterson, and Shawna Marie Hubert for comments on an earlier draft of the manuscript. We also thank Dr. Janet Fulton of Hy-Line International for her support of this project.

### Funding

The work was supported by undergraduate research funds from the College of Agriculture and Life Sciences at Texas A&M University. Genotyping was supported by in-kind contributions from Hy-Line International. The funders had no role in study design, data collection and analysis, decision to publish, or preparation of the manuscript.

### Grant Disclosures

The following grant information was disclosed by the authors:
College of Agriculture and Life Sciences at Texas A&M University.
Hy-Line International.

### Competing Interests

The authors declare there are no competing interests.

### Author Contributions

- Giridhar Athrey conceived and designed the experiments, analyzed the data, contributed reagents/materials/analysis tools, prepared figures and/or tables, authored or reviewed drafts of the paper, approved the final draft.
- Nikolas Faust performed the experiments, analyzed the data, prepared figures and/or tables, approved the final draft.

- Anne-Sophie Charlotte Hieke performed the experiments, analyzed the data, contributed reagents/materials/analysis tools, prepared figures and/or tables, authored or reviewed drafts of the paper, approved the final draft.
- I Lehr Brisbin authored or reviewed drafts of the paper, approved the final draft, provided unpublished data and history of captive populations, commented on manuscript draft.

## Animal Ethics

The following information was supplied relating to ethical approvals (i.e., approving body and any reference numbers):

Sampling of birds were done in accordance with protocols approved by the Texas A&M Institutional Animal Care and Use Committee (IACUC 2016-0065).

## Data Availability

All the raw genotype data is available as Supplemental Files. Raw data is also available on Figshare: Athrey, Giri (2018): Red Junglefowl captive population genetic data. figshare. Dataset. https://doi.org/10.6084/m9.figshare.7081283.v1.

## Supplemental Information

Supplemental information for this article can be found online at http://dx.doi.org/10.7717/peerj.5803#supplemental-information.

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
