# Peer review of "Effective population sizes and adaptive genetic variation in a captive bird population"

_PeerJ, doi:10.7717/peerj.5803_

## Round 0.1 · original submission · Major Revisions

Both reviewers have raised important issues about this manuscript. Please address all of their comments. Also, both have suggested additional analyses that would strengthen this manuscript; please incorporate these where possible or explain why this is not feasible.

There is little discussion of the limitations of this study but this would be a useful inclusion. For example, the microsatellite estimates are based on 15 markers, a rather small panel. Further, the variability of these markers is exceptionally low for microsatellites. Please consider how the choice of markers affects your results. Papers such as Antao et al. 2010 (https://doi.org/10.1111/j.1752-4571.2010.00150.x) may be of use here.

On page 15, you state, "...we believe that the measures of
genomic diversity based on the SNP panel to be more informative about genome-wide patterns of diversity, while the microsatellite-based estimates are consistent with and relevant to our understanding of neutral diversity." This sounds as if you consider "genome-wide patters of diversity" to be opposite to "neutral diversity". Perhaps reword this? Clearly, microsatellites may reflect both genome-side and neutral diveristy.

Please ensure the details of your microsatellites (missing from supporting information) are included in the next version of your manuscript and include number of alleles for each locus. On page 16 you state, "If high microsatellite variability does not correlate positively with genome-wide diversity, then management decisions need to consider the potential limitations of such data." While this is a true statement, one does wonder if the panel you are using is indeed representative of microsatellites across the genome.

Please include a citation for this statement on page 11, "...linkage of loci over a single chromosome can be expected to be stabler over evolutionary time, compared to markers sampled from across the genome."

Finally, in the legend of Table 1, please add "type" after "For each marker". Also Nae is defined in the legend but not included in the table (Ne used as a column header instead).

Reviewer 1 ·

Basic reporting

This manuscript is well written and meets the PeerJ requirements for basic reporting.

One suggestion:
Page 6- the description of the introduction is clear and well described, but it might helpful for the reader for this description to be also supported by a flow diagram, to summarising the population history, estimated population size, and generations in captivity?

Experimental design

The experiment is well defined and executed. I have a few requests for more detail about the methodology, but the approach and implementation is sound.

Page 8- “Longmire Buffer” – can you please provide a reference or supplier for this buffer, or clarify its constituents.

Page 8- “For microsatellite genotyping, a total of 18 microsatellite markers were randomly selected”. Please clarify this statement. Randomly selected from a suite of previously characterized chicken markers? Wouldn’t it have been better to strategically select unlinked markers from across the genome to ensure an unbiased representation of the genome? Please justify why you chose this approach to select markers. I could not find Table S1 that had the list of microsatellite markers- referred to on line 1 of page 8? Please check if this has been accidentally omitted during submission? If it has not already been done, please include the physical location in the genome of all markers in Table S1. You later state on page 9 that “microsatellite loci came from multiple chromosomes”- it would be better to know exactly where each marker is.

Page 9 first paragraph, please include the details of your PCR reaction conditions and PCR cycling conditions. There is currently insufficient detail to replicate the exact methodology.

Page 9- please clarify if the microsatellite markers were amplified singly and then pooled for fragment length analysis (I think this is what you mean); as opposed to being multiplexed, which means many loci amplified simultaneously in a single PCR. If the latter, please provide multiplex PCR conditions. You mention this later, but it would be helpful to also have this made clear early on.

Page 11- Analysis of linked MHC markers. Please clarify if the MHC SNPs were analysed as a single haplotype or as individual SNPs? The former approach would be appropriate, but the latter approach clearly violates the assumption that markers are unlinked.

Validity of the findings

In general the results are well described and supported by the data, but there are a few instances where this is less true. The authors could improve the following areas:

Page 13- “ranging from 0.359 to 0.371 to 0.379” consider re-phrasing this sentence, it makes it seem as if you only genotyped 3 loci, which is not what you did. Your meaning is not clear here.

Page 13- “The Ne estimates were in the low single digits for the microsatellite marker dataset, except for the heterozygote excess method.” This sounds too colloquial, please re-phrase to be more specific.

Page 14,-add “As expected”... analysis of linkage disequilibrium based on the SNP loci on chromosome 16 revealed very high D’ values across all pairwise comparisons. They are all physically linked, and thus should show high LD?

Page 15- first line “SNP loci.” Specify, that the SNP loci are MHC linked SNPs – otherwise when read in isolation a reader could mis-interpret this statement and assume they are genome-wide unlinked SNPs.

Page 16- I disagree with the statement “While it is difficult in our study to determine how the SNP loci may be representative of whole genome diversity, there is no particular reason to believe that chromosome 16 would be experiencing selection or drift in ways that are inconsistent with the rest of the genome.” MHC loci could very easily be experiencing differential selection strengths or different modes of selection than the genome as a whole, particularly if the loci are all located on a microchromosome. Please clarify why you think there is no difference? Without further explanation, I don’t think the following discussion statement if supported by the data: “Therefore, we believe that the measures of genomic diversity based on the SNP panel to be more informative about genome-wide patterns of diversity, while the microsatellite-based estimates are consistent with and relevant to our understanding of neutral diversity.” Maybe the authors should consider a discussion of whether it is more important to manage neutral variation or genetic diversity of regions that may underpin adaptive traits?

Page 16- again I don’t think your data support this statement: “If high microsatellite variability does not correlate positively with genome-wide diversity…”. Clarify if this is intended to be speculation or a conclusion.

Page19 – “We found that the demographic history corresponds to a great extent with the genetic estimates.”.. but different Ne estimators were reflecting different historical demographic events? Please clarify?

Additional comments

In general the discussion is a bit long, and it is not obvious what the main result is and why it is important? Can the authors please consider streamlining the discussion in general (reduce word count) and more clearly articulating the main result in the first paragraph of the discussion.

I have a few other suggestions for additional analysis or interpretation that the authors could consider to improve the manuscript:

Page 17 – I think this is very interesting: “Therefore, for both the LD and MC methods, we believe that the estimates obtained here represent the founding event of the Texas population in 2016.” And might be what makes your study unusual, you can comment on the temporal uncertainty of Ne estimators, and that often it isn’t really clear WHEN the estimate generated is relevant to? You could make more of this point and compare what each estimator is doing, what are their strengths and weaknesses? It also means you can highlight how you are not comparing apples to apples when comparing “Ne” using different methods, if they are taking different temporal snapshots of the populations? Can we use this knowledge to our advantage and can you suggest a strategy to get a better picture of population history in terms of Ne? If possible, this would increase the impact and utility of your findings. Could you incorporate this information in the flow chart I suggested earlier about the population history? You are in a fairly unique position having genotyped 100% of the population and knowing so much about the population history, please make the most of this!

Page 18- If you want to talk about the impact of domestication, you will need to include a new analysis comparing MHC SNP variation in your Red Jungle fowl to that of domestic chicken strain, specifically. This could be a nice addition, and there may be sufficient publicly available chicken data (at least from the published genome if not more) to implement this.

Minor typographical errors:
Page 18- remove double reference to Wright “Wright et al. (Wright et al., 2010)”

Supplemental files:
Table S1 with full list of primers appears to be missing?
Typo in the genotypes file, in the tab name: “Microsatellite Genptypes”

Reviewer 2 ·

Basic reporting

This manuscript is well written with professional English.

Experimental design

Microsatellite information
Manuscript stated that microsatellites would be listed with their primer sequences in S1, but this is not the case. It is important for reproducibility that primer names are listed and primer sequences and references included. The genomic location/coordinates of the microsatellite markers will also be very informative to assess the genomic coverage of this study's genetic diversity. I strongly recommend the authors include microsatellite names, references, and genomic locations.

Using a chr16 specific SNP chip
Focusing on chr16 (SNPs only present on chromosome 16) introduces certain limitations and biases to the study. Chromosome 16 contains the MHC, which is mentioned in the manuscript, but the consequences of this is not adequately appreciated. The MHC is often a target of natural selection, as it has a crucial role in pathogen recognition and immune surveillance, and it's diversity is shaped by host-pathogen co-evolution. The MHC often exhibits strong linkage disequilibrium, so the results herein of strong linkage among SNP markers is not at all surprising, given the proximity to the MHC. It is also not surprising given the historically small population sizes of founders and indeed the small founder size of the studied population, as mentioned by the authors. Nevertheless, it would be extremely difficult in this population to disentangle the linkage observed to be as a result of founder effect, small population size, or positive selection. For example, a historical exposure to disease could result in positive selection for one particular protective MHC haplotype, greatly increasing its frequency in the historical population, and consequently resulting in a major haplotype in this subsequent population. It is therefore difficult to directly compare results from microsatellites to those from these SNPs. I would believe that the microsatellite markers provide a better insight into the wider genomic trend, while the chr16 SNPs provide an insight into the immunogenetic diversity present in the population.

The authors could use the SNP information from this population to compare to SNP diversity present in other chicken populations (domestic chickens, hyrbid RJF/chicken, pure RJF) - much data are available online. Understanding the relationship between the chr16 in this population and that in others could be extremely informative. For example, is the major MHC haplotype on chr16 associated with resistance to any particular poultry disease?

I would also be interested in seeing a generation by generation breakdown of microsatellite and haplotype frequencies. Indeed, the genotypic information in the founders provides a clear insight into the founding genetics.

Validity of the findings

As mentioned in the previous section, the consequences of focusing on a region including the MHC requires much greater appreciation and discussion in the manuscript. Indeed, I believe that the following sentence in the discussion: "there is no particular reason to believe that chromosome 16 would be experiencing selection or drift in ways that are inconsistent with the rest of the genome" is wrong in every way possible. There is EVERY reason to believe that chr16 in particular would experience selection in ways different to the genome. MHC can be shaped by pathogen-mediated selection and also can be shaped by MHC-driven mate choice. Authors should make themselves aware of the selective forces that influence the MHC and therefore will influence chicken chrs16, which is likely to change interpretation of results.

In the conclusion, authors state that they have "showed the dynamics of genetic diversity and effective population sizes" - I would argue they have not investigated any dynamics. Authors have assessed metrics of genetic diversity and effectively population size in one single population at one given time point. This does not represent dynamics. Dynamics could be inferred using multiple time points across populations of varying size but such data that is not available in this study.

Additional comments

This paper is well-written but requires further work to appropriately and accurately interpret results. It is particularly difficult to accept that chr16 is a neutrally evolving genomic region, since decades of research in the MHC across multiple taxa consistently find the MHC is affected by selection. In many studies, MHC is compared to microsatellite markers in order to capture different evolutionary forces that impact genetic diversity in populations. This study would greatly benefit from the use of genome-wide SNP panels, but in the absence of this, I believe a better integration of previous results and publicly accessible genotype data of other chicken populations will greatly improve this manuscript.

---

## Round 0.2 · Minor Revisions

We appreciate your careful consideration of reviewer comments and find the manuscript greatly improved.

There are several points that remain in need of correction/clarification:

Line 31 Suggest adding “many” before “captive lab”. Also, change “analysis” to “analyses” for verb agreement (and accuracy!)
Line 33 I’m not entirely sure why these are surprising – clarify?
Lines 33-34 Hedrick 1992 is listed twice – I note that this occurs throughout the manuscript. Please amend.
Line 45 This sentence is a bit awkward to read. Perhaps remove “and intended”
Lines 125-126 Please add this information regarding current population size to Figure 1
Lines 138-142 Very long sentence. Please break at “due to our..”
Lines 215-217 Please change information in parentheses to the abbreviation (e.g. “HE” rather than “Heterozygote excess”) and include the abbreviation “LD” after the last method is listed.
Line 219 Suggest adding “artificial” in front of “selection” because surely natural selection marches on!
Line 224 Remove space before full stop
Line 269 “0.98” should be “0.098”
Line 278 Please rearrange to read, “…high D’ (linkage disequilibrium) values across off pairwise comparisons. D’ was estimated…”
Line 280 Suggest inserting “of a small founding population” after “captive breeding”
Lines 289-293 Please include the methods for these results in “Methods”
Line 297 Insert “(Table 3)” after “excess method”
Line 300 Insert “number of individuals in the” after “total”
Lines 304-311 This paragraph does not report results and is better suited to the discussion. Integrate it with the comments already included there.
Lines 322-325 I think you can (and should) make this stronger – it doesn’t just “raise questions” but suggests that assessing such populations with msats is prone to underestimation of functional diversity and effective population size.
Line 345 Please amend to, “…with a history of small population size” [size=singular]. Also, suggest changing “the predominant” to “a predominant”
Line 358 Suggest changing “takeaway” to “conclusion”
Line 366 Remove “whole-genome” – this study only looked at one chromosome so this sound a bit misleading.
Line 373 Suggestion changing “assumptions” to “conditions” because estimates don’t arise from assumptions
Line 373-377 It’s a bit funny to talk about sample size in the context of your study since you have typed the whole population. Suggest revising to, “…biased downward due to the small number of individuals in this study, but such sample sizes are not unusual in studies of endangered populations. Several of the Ne estimators are predicted to result in biased estimates at sample sizes less than N=50. However, in this study, 100% of individuals were sampled.”
Line 387 Why is “HE” at the end of this sentence?
Line 391-393 This sentence repeats what is said in lines 386-387
Line 417 Suggest replacing “came” with “are likely to have come”
Lines 444-447 Suggest replacing this sentence with, “While we cannot distinguish between the effects of directional selection and bottleneck in our study, the latter is expected given the known history of this population. The major shift in diversity from wild to captive populations within 60 generations in intriguing, especially in the context of domestication selection.”
Lines 452-453 Suggest changing to, “Given this history, the low effective population sizes estimates we found are unsurprising.” Or something along these lines
Figure 1 Please explain in the legend what the colours denote. Also, please check formatting of the figure for spacing, places where text is cut off, etc.
Table 1 Please put the full (not rounded) figures in this table so that it matches the values given in the body of the manuscript.
Table S1 Please change heading of column 2 to “Number of alleles…” and under the “Notes” column, give the reason why those three markers were removed.

---

## Round 0.3 · accepted · Accept

Thank you for your careful attention to comments and for your revisions.

#